# Research on the Microstructure, Mechanical Properties and Strengthening Mechanism of Nanocrystalline Al-Mo Alloy Films

**DOI:** 10.3390/nano14241990

**Published:** 2024-12-12

**Authors:** Ying Wang, Huanqing Xu, Yulan Chen, Xiaoben Qi, Ning Zhong

**Affiliations:** 1School of Materials Science, Shanghai Dianji University, Shanghai 201306, China; ywang@sdju.edu.cn (Y.W.); hh369792@126.com (H.X.); 15933618107@163.com (Y.C.); 2College of Ocean Science and Engineering, Shanghai Maritime University, Shanghai 201306, China; ningzhong@shmtu.edu.cn

**Keywords:** nanocrystalline, Al-Mo alloy film, microstructure, mechanical properties, strengthening mechanism

## Abstract

In this work, the Al-Mo nanocrystalline alloy films with Mo contents ranging from 0–10.5 at.% were prepared via magnetron co-sputtering technology. The composition and microstructure of alloy thin films were studied using XRD, TEM, and EDS. The mechanical behaviors were tested through nanoindentation. The weights of each strengthening factor were calculated and the strengthening mechanism of alloy thin films was revealed. The results indicate that a portion of Mo atoms exist in the Al lattice, forming a solid solution of Mo in Al. The other part of Mo atoms tends to segregate at the grain boundaries, and this segregation becomes more pronounced with an increase in Mo content. There are no compounds or second phases present in any alloy films. As the Mo element content increases, the grain size of the alloy films gradually decreases. The hardness of pure aluminum film is 2.2 GPa. The hardness increases with an increase in Mo content. When the Mo content is 10.5 at.%, The hardness of the film increases to a maximum value of 4.9 GPa. The fine grain ($∆H_{gb}$), solid solution ($∆H_{ss}$), and nanocrystalline solute pinning ($∆H_{nc,ss}$) are the three main reasons for the increase in the hardness of alloy thin films. The contribution of $∆H_{gb}$ is the largest, accounting for over 60% of the total, while the contribution of $∆H_{ss}$ accounts for about 30%, ranking second. The rest of the increase is due to $∆H_{nc,ss}$.

## 1. Introduction

Due to their excellent performance, nanocrystalline materials have been increasingly applied in fields such as aerospace and electronic information. Their application has always been a hot research topic in the academic community [1,2,3,4]. When the grain size decreases to the nanometer level, the volume fraction of the grain boundaries will increase exponentially. The stability of nanocrystalline material systems will be significantly reduced, making it easy for spontaneous grain growth to occur. For example, pure nanocrystalline metals such as Cu [5], Ni [6], and Al [7] will exhibit grain growth under external loading or at very low heating temperatures. Some nanocrystalline pure metal grains may even coarsen at room temperature [8]. The instability of their structure limits the application of nanocrystalline pure metals.

Therefore, improving the stability of nanocrystalline metals has always been an important research direction [9,10,11]. Some studies have shown that adding alloying elements to pure nanocrystalline metal is an effective way to improve their stability. Devaraj [12] studied the effect of the addition of Mg on the stability of nanocrystalline Al. It was found that, after annealing at 300 °C for 3 h, the grain size of the pure Al film doubled, while the size of the Al-10 at.% Mg alloy film only increased by 69%. Rajagopalan et al. [13] added Ta to nanocrystalline Cu. The grain size of the Cu-Ta nanocrystalline alloy did not show significant growth even after annealing at 400 °C. In addition, similar research results were reported for the Cu-Zr [14], FeNi-Zr [15], and Ni-W [16] systems. These additional alloying elements can limit the coarsening of these grains by reducing the system’s free energy and pinning the nanocrystalline boundaries, thereby improving stability [17]. Therefore, compared to pure metals, nanocrystalline alloys have more practical value. It is also of greater research significance to study their mechanical behavior and corresponding reinforcement mechanism.

However, the strengthening mechanism of nanocrystalline alloys is also more complex [18,19,20]. With the change in alloy element content, the grain size, solute content, and properties of the grain boundaries of the nanocrystalline alloys will undergo corresponding changes. The interweaving of these microstructural factors makes it extremely difficult to reveal the strengthening mechanism [21]. Distinguishing the roles of various structural factors and studying them separately is an effective way to reveal the strengthening mechanism of nanocrystalline alloys. Some studies [20,22,23] have already attempted to do this through theoretical calculations. However, most of these studies obtained nanocrystalline alloys with fixed compositions and calculated the effects of various strengthening factors. Few systematic studies have been conducted on a series of nanocrystalline alloys to investigate the changing trends and weights of various strengthening factors during the alloy content variation process.

In this study, Al-Mo nanocrystalline alloy films were synthesized via magnetron sputtering technology. The effect of different molybdenum element contents on the microstructure and comprehensive mechanical properties of the alloy systems was studied in detail. Subsequently, the roles of various strengthening factors in the alloy films were calculated. The strengthening mechanism of nanocrystalline Al-Mo alloy films is revealed, based on the effects of various microstructural factors.

## 2. Materials and Methods

### 2.1. Film Deposition

The Al-Mo nanocrystalline alloy films with a Mo content ranging from 0 to 10.5 at.% were synthesized via multi-target magnetron co-sputtering technology using an ANAVA SPC-350 magnetron sputtering instrument (Anelva, Tokyo, Japan). The Al and Mo targets with a diameter of 75 mm were controlled by two RF cathodes, respectively. The distance between the target material and the substrate was 50 mm. The single-crystal Si wafers with dimensions of 15 mm × 15 mm were cleaned with alcohol ultrasonic for 15 min and dried before being placed on substrate racks in vacuum chambers. After evacuating the chamber to 5 × 10^−4^ Pa, the high-purity Ar gas, with a purity of 99.999%, was introduced. During the sputtering process, the working pressure was maintained at 6 × 10^−1^ Pa. A series of Al-Mo alloy thin films with a Mo content ranging from 0 to 10.5 at.% were prepared by changing the Mo target power while keeping the Al target power at 200 W. The Mo target powers were 0, 10, 20, 30, 40, and 50 W, respectively. The substrate was not heated or subjected to negative bias throughout the entire deposition process. The total deposition time was controlled at 2 h. The thickness of all alloy films was controlled at around 2 μm.

### 2.2. Films Characterization

The elemental contents (Al, Mo) of the alloy films were measured using an EDS (EDX, Oxford instruments INCA) (Oxford Instruments, Oxford, UK) attached to the scanning electron microscope (S-3400N SEM, Hitachi, Tokyo, Japan). The phase structure of the Al-Mo nanocrystalline alloy films was determined via X-ray diffraction (XRD) with Cu–Kα radiation using a D8 X-ray diffractometer (Bruker, Karlsruhe, Germany). The scanning range of 2θ was 30–70°, with a scanning speed of 5°/min. The microstructure was characterized using a JEOL-2100F TEM microscope (JEOL, Tokyo, Japan). The TEM samples were prepared using the soluble salt substrate method [24]. The comprehensive mechanical properties of the Al-Mo alloy systems were determined using a Step 300-NTH3 nano indentation (Anton Paar, Graz, Austria). The maximum load applied was 10 mN. The loading and unloading speed was 30 mN/min. The holding time was 10 s. The hardness and elastic modulus information of the films was analyzed using the Oliver–Pharr method [25]. To ensure the validity of the data, all samples were measured at more than 20 points.

## 3. Results and Discussions

### 3.1. The Microstructure of Alloy Films

Figure 1 presents the XRD pattern of the Al-Mo alloy films. As shown in the figure, all films contained a set of FCC Al peaks and no other Al compound peaks were found. This indicates that the film formed an Al-Mo solid solution. The XRD pattern of the pure Al film and the Al-0.9 at.% Mo film shows sharp diffraction peaks of Al (111) and Al (200), as well as weak diffraction peaks of Al (220). As the Mo content further increases, the Al (111) peak gradually widens while the Al (200) peak gradually weakens, indicating a gradual decrease in grain size. This is because the continuous addition of Mo atoms causes a large amount of lattice distortion in the Al film, which destroys the integrity of the Al grains. The diffraction peaks of Al (200) and Al (220) disappeared after the Mo element content increased to 10.5 at.% while the Al (111) peaks also became broadened and diffused. This indicates that the Al-Mo alloy film contains nanocrystalline or amorphous structures. Additionally, it is worth noting that as the Mo content increases, all diffraction peaks shifted towards larger angles. This is due to the substitution of larger-radius Al atoms with smaller-radius Mo atoms, resulting in the contraction of the Al lattice.

According to the XRD results, the films’ grain size and interplanar spacing were calculated using Scherrer formula and Bragg’s law, respectively. In Figure 2a, the film grain size continues to decrease while the Mo content increases. In pure Al film, the grain size is about 121 nm. When the Mo element content increases to 0.9 at.%, it rapidly decreases to about 84 nm. Further increasing the Mo content leads to a continuous decrease in the grain size of the alloy film. When the Mo element content reaches 10.5 at.%, the grain size of the alloy film decreases to about 19 nm. This is because the high non-equilibrium of magnetron sputtering causes the sputtered particles to quickly lose kinetic energy when deposited on the substrate. The limitations of the dynamic conditions force Mo atoms to remain in the Al lattice. The addition of the Mo atoms causes severe distortion of the Al lattice, resulting in a decrease in grain size. In Figure 2b, the films’ interplanar spacing continuously decreases with an increase in Mo contents. The dashed line represents the fitting curve.

In order to further determine the microstructure of the Al-Mo alloy films, a plane observation of the alloy films was performed using TEM. The results are shown in Figure 3. According to the grain size distribution diagrams in the lower right corner of Figure 3a–c, the grain size was about 120 nm in the pure Al film. The grain size continued to decrease with the increase in Mo. In the Al-7.8 at.% Mo alloy film, the grain size was only a dozen nanometers. In addition, the HRTEM analysis indicated that the interplanar spacing of the Al lattice gradually decreases with the increase in Mo content. This trend is consistent with the results of the XRD analysis.

The relationship between Mo content within the grain and the global Mo content of films is shown in Figure 4. The global Mo content is measured via EDS. The grain interior Mo content is calculated using Formula (1) [26]:(1)$$a=G\left\{2\left[cR_{1}\left(1\right)+\left(1-c\right)R_{2}\left(1\right)\right]-\beta \mathrm{lg}\left[cn_{1}^{C}+\left(1-c\right)n_{2}^{C}\right]\right\}$$
where *a* is the lattice constant shown in Figure 2b. *G* and *E* are constants (for FCC structure, $G=\sqrt{2}$, $E=\frac{2-\sqrt{2}}{2}$); *A* and *B* are the numbers of the shortest and second shortest bonds, respectively. $R_{1}\left(1\right)$ and $R_{2}\left(1\right)$ are half of the bond length. $n_{1}^{C}$ and $n_{2}^{C}$ are the numbers of covalent electrons.

In Figure 4, the dashed line shows that the grain interior’s Mo content is equal to the global Mo content. As the global Mo content increases, the grain interior’s Mo content also shows an upward trend, but is always generally lower than the global Mo content. This indicates the phenomenon of Mo atom segregation at grain boundaries in Al-Mo alloy films. When the total Mo content is below 2.1 at.%, this difference is not significant, but when the Mo content is above 5.3 at.%, this trend gradually increases.

### 3.2. Mechanical Behaviors of Alloy Films

Figure 5 shows the nanoindentation measurement results of the Al-Mo alloy film. As shown in Figure 5a, the change in hardness can be divided into two stages with the increase in Mo content. When the Mo content is in the range of 0–2.1 at.%, the hardness rapidly increases from 2.2 GPa to 3.7 GPa. When the Mo content is over 2.1 at.%, although the hardness continues to increase, the growth rate gradually slows down. When the Mo content increases to 10.5 at.%, the hardness reaches the maximum value of 4.9 GPa. The elastic modulus in Figure 5b shows an almost linear increasing trend with the increase in Mo content. The dashed line is the fitting curve.

There are three strengthening factors that contribute to the increase in the hardness of alloy films. On the one hand, as the Mo content increases, the grain size of the alloy film continues to decrease. The hardness increment of $∆H_{gb}$ caused by grain refinement is one of the important reasons for the increase in the hardness of Al-Mo alloy films.

The value of $∆H_{gb}$ was calculated using Formula (2). In this article, $∆H_{gb}$ represents the difference in hardness between Al-Mo alloy films with different Mo contents and pure Al films.
(2)$$∆H_{gb}=H-H_{Al}=ak(d^{-1/2}-d_{Al}^{-1/2})$$
where a = 3.6 represents the conversion relationship between hardness and strength. The value of k is taken as 3.795 GPa/nm^−1/2^ [18].

On the other hand, solid-solution atoms also have pinning and hindering effects on the movement of dislocations. Therefore, solid-solution strengthening is another important reason for the enhancement of the hardness of Al-Mo alloy thin films. The hardness increment $∆H_{ss}$ can be calculated using the Fleischer model.
(3)$$∆\sigma_{Fleischer}=\sqrt{2}{\left(0.2\right)}^{3/2}G{\left|\varepsilon_{G}^{’}-m\varepsilon_{b}\right|}^{3/2}c^{1/2}$$
where c is the solute content and m is a constant representing the type of dislocation, which is m = 3 for this paper. G is the shear modulus of the alloy film, which can be calculated from the elastic modulus in Figure 5b. $\varepsilon_{b}$ is the lattice mismatch coefficient; $\varepsilon_{b}=\left(db/dc\right)/b$. $\varepsilon_{G}^{’}$ is the modulus mismatch coefficient.
$$\varepsilon_{G}^{’}=\frac{\varepsilon_{G}}{1+0.5\varepsilon_{G}}$$

Finally, Rupert et al. [21] suggested that when the grain size decreases to the nanometer level, the grain boundaries also have a pinning effect on the movement of dislocations. The hardness increment $∆\sigma_{nc,ss}$ caused by the nanocrystalline solution pinning strengthening can be expressed as follows:(4)$$∆\sigma_{nc,ss}=\frac{Gb}{d}\varepsilon_{nc}$$

Based on the above, the total hardness increment $∆H_{Total}$ of the Al-Mo alloy film can be expressed as follows:(5)$$∆H_{Total}=∆H_{gb}+∆H_{ss}+∆H_{nc,\;ss}$$

Figure 6 shows the variation in the curves of $∆H_{Total}$, $∆H_{gb}$, $∆H_{ss}$, and ${∆H}_{nc,ss}$ with the Mo content. As shown in the figure, the increment in hardness caused by the three strengthening factors gradually increases with the increase in Mo content. The contribution of grain refinement ($∆H_{gb}$) is the most significant, exceeding 60%, followed by solid-solution strengthening ($∆H_{ss}$), which accounts for about 30% of the effect, and the contribution of the nanocrystalline solution pinning (${∆H}_{nc,ss}$) is the smallest, comprising only about 10% of the total effect.

The red line in Figure 7 represents the theoretically calculated hardness increment $∆H_{C}$, while the black line represents the experimentally measured hardness increment $∆H_{M}$. Although the $∆H_{M}$ curve is slightly higher than the $∆H_{C}$ curve, the trends of the two are generally similar. This indicates that the consistency between the theoretical calculation and experimental results is good. Although the calculated results are in good agreement with the experimental results, there is still room for discussion regarding this theoretical model. The Hall Petch formula and solid solution strengthening model are both derived from coarse-grained materials, and their applicability at the nanocrystalline scale needs to be discussed. Some studies have shown that when the grain size decreases to 20–30 nm, traditional dislocation models will fail. The nanocrystalline Al alloy will exhibit the inverse Hall–Petch effect and solid solution softening phenomenon. This may also be the reason the theoretical calculated hardness of Al-10.5 at.% Mo films continues to rise, while the measured hardness tends to flatten out.

## 4. Conclusions

In this work, Al-Mo alloy films were prepared using magnetron co-sputtering technology. The effect of different Mo element contents on the microstructure and comprehensive mechanical properties of the alloy films was studied. The strengthening mechanism of the alloy films was revealed. The following main conclusions were obtained:

All the Al-Mo alloy films formed an Al-Mo solid solution. No compounds or second phases appeared. With the increase in Mo, the grain size rapidly refines. The additional Mo atoms exhibited the phenomenon of grain boundary segregation. The higher the Mo content, the more pronounced the trend of grain boundary segregation.The hardness and elastic modulus of the Al-Mo nanocrystalline alloy films gradually increased with the increase in Mo content. The fine grain ($∆H_{gb}$), solid solution ($∆H_{ss}$), and nanocrystalline solute pinning ($∆H_{nc,ss}$) are the three main reasons for the increase in the hardness of alloy thin films. Among them, the contribution of $∆H_{gb}$ is the largest, accounting for over 60% of the total response, while the $∆H_{ss}$ accounts for about 30%, ranking second. The rest is due to the contribution of $∆H_{nc,ss}$.

## Figures and Tables

**Figure 1 nanomaterials-14-01990-f001:**
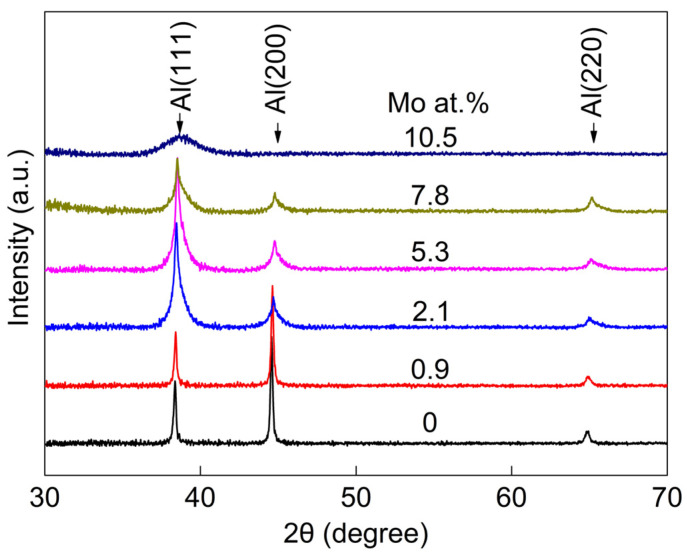
The XRD analysis results of Al-Mo alloy thin films.

**Figure 2 nanomaterials-14-01990-f002:**
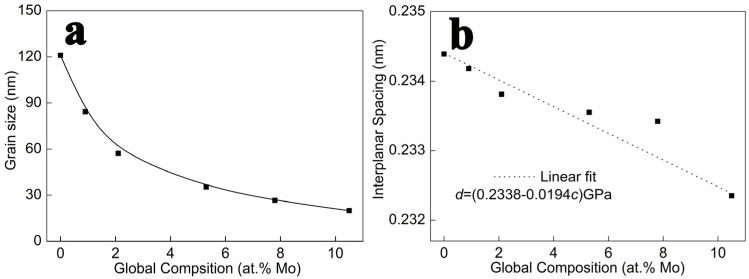
The relationship between grain size (**a**) and the interplanar spacing (**b**) of alloy thin films and Mo content.

**Figure 3 nanomaterials-14-01990-f003:**
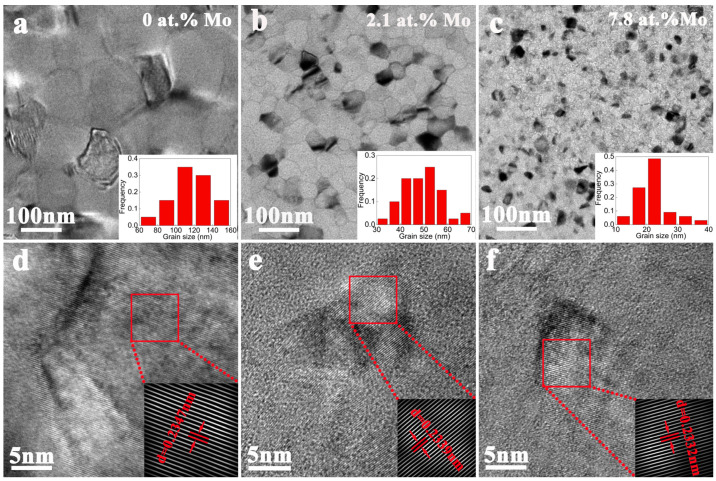
TEM bright field image and HRTEM image of pure Al (**a**,**d**), Al-2.1 at.% Mo (**b**,**e**), and Al-7.8 at.% Mo (**c**,**f**) alloy film.

**Figure 4 nanomaterials-14-01990-f004:**
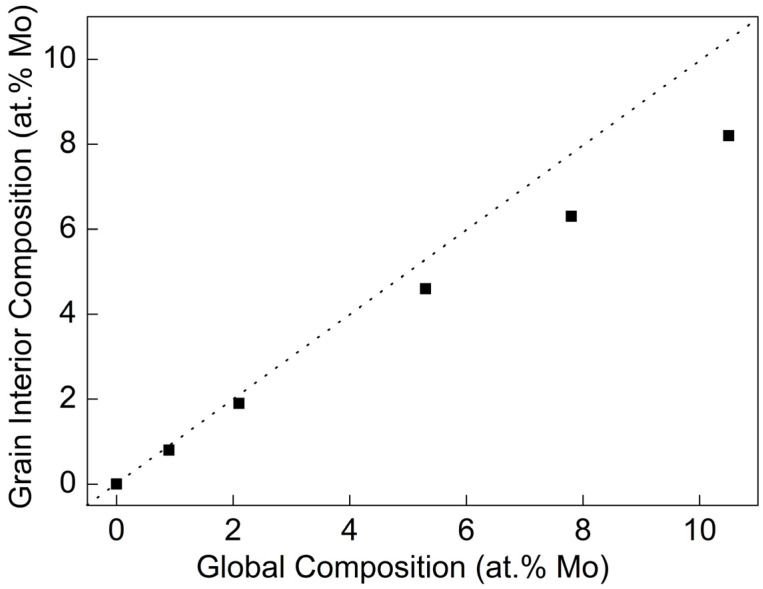
The variations in the grain interior Mo content and global Mo content of Al-Mo films.

**Figure 5 nanomaterials-14-01990-f005:**
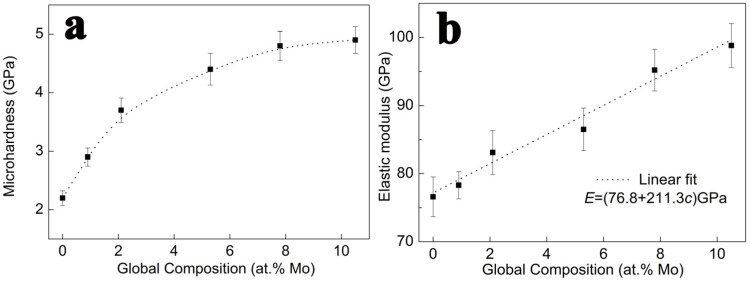
The nanoindentation measurement results of Al-Mo alloy film’s (**a**) hardness and (**b**) elastic modulus.

**Figure 6 nanomaterials-14-01990-f006:**
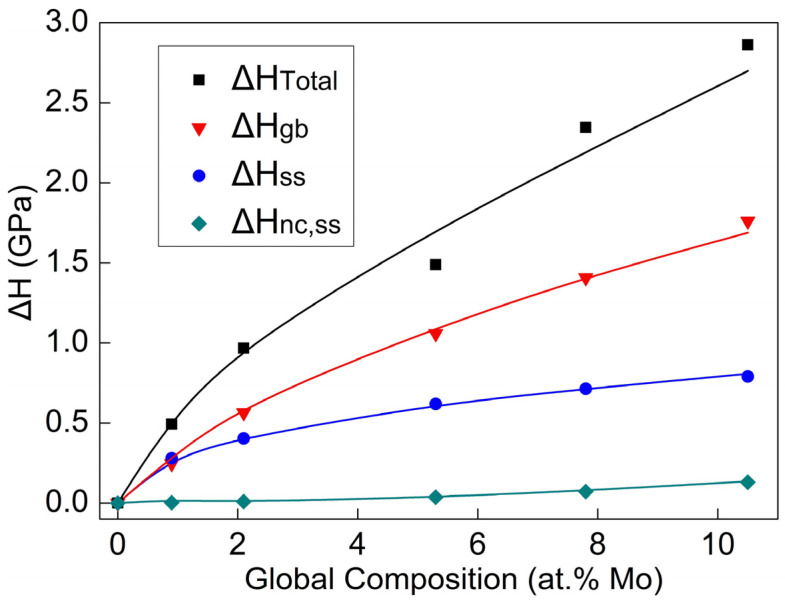
The variation curves of $∆H_{Total}$, $∆H_{gb}$, $∆H_{ss}$, and ${∆H}_{nc,ss}$ with Mo content.

**Figure 7 nanomaterials-14-01990-f007:**
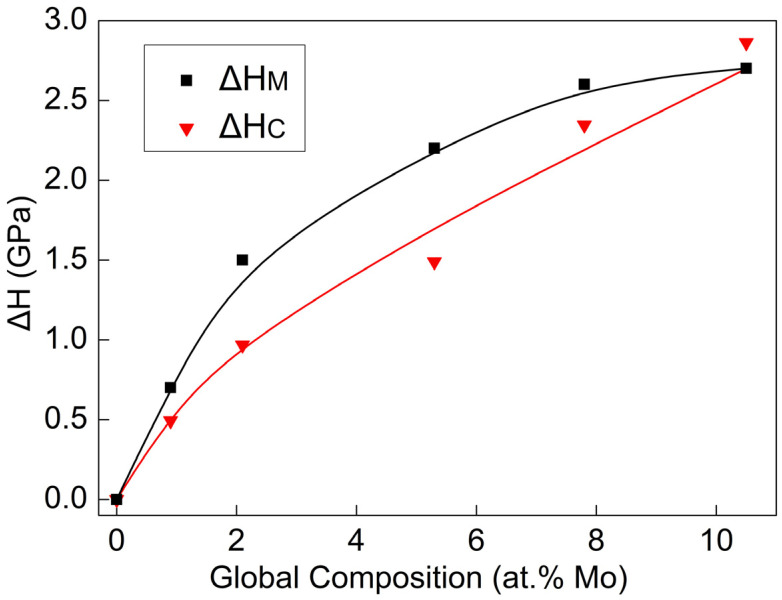
The calculated and measured hardness increment of alloy films.

## Data Availability

The authors confirm that the data supporting the findings of this study are available within the article.

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
