# Peer review of "Research on the Microstructure, Mechanical Properties and Strengthening Mechanism of Nanocrystalline Al-Mo Alloy Films"

_nanomaterials, 2024, doi:10.3390/nano14241990_

Round 1
Reviewer 1 Report
Comments and Suggestions for Authors
The manuscript delves thoroughly and methodically into the strengthening mechanisms in Al-Mo nanocrystalline alloy films prepared by magnetron co-sputtering. The authors detail the clear link between microstructure, particularly grain size and solute content, and mechanical properties such as hardness and elastic modulus. The manuscript is well-written and organized, moving precisely from material preparation to the discussion of results. Nevertheless, a few dimensions must be incorporated into the study for further depth and broader applicability. Minor revisions before acceptance are recommended.
1- The study quantifies the contributions from several strengthening mechanisms (grain refinement, solid solution, nanocrystalline solute pinning), which is excellent. The drawback is that the novelty is incremental since these mechanisms have been well-studied in nanocrystalline systems. The manuscript would benefit from a more direct comparison with other alloy systems or with previous studies on Al-based alloys to further establish its unique contribution.
2- The thermal stability of these nanocrystalline materials is very much in question regarding their possible practical application. Although some material is provided on the grain boundary segregation of Mo, no thermal stability tests or other such experiments are included (for example, annealing studies). Determining either grain growth or retaining some mechanical properties at elevated temperatures would considerably impact the manuscript.
3- High hardness is often associated with low ductility and brittleness. The manuscript should cover this trade-off, at least experimentally or by discussing expected behavior concerning existing literature.
4- Although the mechanical properties of the Al-Mo alloy films are sufficiently discussed, practical applications of these films in specific areas (such as aerospace microelectronics) are avoided. This could give the study a broader appeal.
5- 0-10.5 at.% Mo is a reasonable selection, but further explanation to justify the choice of this range (e.g., specific applications or previous findings) would be helpful.
6- Lattice strain due to the calculated diffraction peaks could be optimized since progress found in purity enhancement with increasing Mo content is visible in the XRD results.
7- Although these contributions to hardening are well calculated and clearly match the experimental data, a brief discussion about the accuracy and limitations of the applied models would be good.
8- More details should be provided about preparing samples for TEM analysis. Was FIB resource employed, or did it involve other means? It is essential information in obtaining fine-quality images.
9- The document cites relevant literature, but including recent studies on thermal stability in nanocrystalline alloys or other Al, systems would bring a more current context into play.
10- The resolution of the images could be enhanced to visualize grain size better.
11- It may be helpful to provide a quantitative assessment of the grain size observed in TEM images compared to the grain size generally simulated from X-ray diffraction.
12- Enhance the contrast in the TEM images to make grain structures and boundaries more distinguishable.
13- The effects of deposition parameters, such as working pressure, power, or substrate bias, on grain size must not be cheaper. They could affect the nanocrystalline structure significantly.
14- In the case of magnetron sputtering, impurities from both the chamber and the targets can come into the system. Were some purity analyses done or steps taken to reduce contamination in the targets mentioned for these cases?
15- Additional details, such as the deposition rate and process duration, are missing.
Author Response
Comments 1: The manuscript delves thoroughly and methodically into the strengthening mechanisms in Al-Mo nanocrystalline alloy films prepared by magnetron co-sputtering. The authors detail the clear link between microstructure, particularly grain size and solute content, and mechanical properties such as hardness and elastic modulus. The manuscript is well-written and organized, moving precisely from material preparation to the discussion of results. Nevertheless, a few dimensions must be incorporated into the study for further depth and broader applicability. Minor revisions before acceptance are recommended.
Response 1:
Thank you for your positive evaluation of the manuscript.
Comments 2: The study quantifies the contributions from several strengthening mechanisms (grain refinement, solid solution, nanocrystalline solute pinning), which is excellent. The drawback is that the novelty is incremental since these mechanisms have been well-studied in nanocrystalline systems. The manuscript would benefit from a more direct comparison with other alloy systems or with previous studies on Al-based alloys to further establish its unique contribution.
Response 2:
Thank you for the reviewer's suggestion. We have added relevant descriptions in the introduction.
Comments 3: The thermal stability of these nanocrystalline materials is very much in question regarding their possible practical application. Although some material is provided on the grain boundary segregation of Mo, no thermal stability tests or other such experiments are included (for example, annealing studies). Determining either grain growth or retaining some mechanical properties at elevated temperatures would considerably impact the manuscript.
Response 3:
Thank you for the reviewer's comments. The stability is indeed an important property of nanocrystalline alloys. In fact, we have also conducted some research on the stability of Al-Mo nanocrystalline alloys and achieved some results, as shown in figure 1. However, considering that stability is not the focus of this article, we will focus on the stability of nanocrystalline Al-Mo alloys in the next article.
Fig.1 XRD patterns of Al-7.8 at.% Mo and Al-12.7 at.% Mo alloy films annealed at different temperatures
Comments 4: High hardness is often associated with low ductility and brittleness. The manuscript should cover this trade-off, at least experimentally or by discussing expected behavior concerning existing literature.
Response 4:
Thank you for the reviewer's suggestion. As you mentioned, nanocrystalline alloys often lose some plasticity and toughness while achieving high strength and hardness, which has been proven by numerous studies. The indentation depth-load curve of nanoindentation in this article can also illustrate this issue, as shown in Figure 2. Under the same conditions, the higher the Mo content, the shorter the plateau of the load holding stage on the curve, indicating a decrease in the plasticity of the film.
Fig.2 The indentation depth-load curve of Al-Mo alloy film
Comments 5: Although the mechanical properties of the Al-Mo alloy films are sufficiently discussed, practical applications of these films in specific areas (such as aerospace microelectronics) are avoided. This could give the study a broader appeal.
Response 5:
Thank you for the reviewer's suggestion. We have added relevant descriptions of the application field of nanocrystalline in the introduction.
Comments 6: 0-10.5 at.% Mo is a reasonable selection, but further explanation to justify the choice of this range (e.g., specific applications or previous findings) would be helpful.
Response 6:
Thank you for the reviewer's suggestions. Based on the research results of this article, when the Mo content is below 10.5 at.%, the film forms a solid solution of Mo in Al. This simple structure is more suitable for studying and analyzing reinforcement mechanisms. When the Mo content exceeds 10.5 at.%, the film forms an amorphous structure, which exceeds the research scope of this paper.
Comments 7: Lattice strain due to the calculated diffraction peaks could be optimized since progress found in purity enhancement with increasing Mo content is visible in the XRD results.
Response 7:
Thank you for the reviewer's suggestion. As you mentioned, the shift in XRD diffraction peak position indicates an increase in Mo content. This article is based on this result to obtain the interplanar spacing of different Al Mo alloy films. Furthermore, the intragranular Mo content was calculated using formula 1. Finally, the effect of solid solution strengthening was calculated.
Comments 8: Although these contributions to hardening are well calculated and clearly match the experimental data, a brief discussion about the accuracy and limitations of the applied models would be good.
Response 8:
Thank you for the reviewer's suggestions. We have added relevant discussions in the discussion section of the revised manuscript.
Comments 9: More details should be provided about preparing samples for TEM analysis. Was FIB resource employed, or did it involve other means? It is essential information in obtaining fine-quality images.
Response 9:
Thank you for the reviewer's comments. The TEM in the paper was prepared using ion thinning equipment instead of FIB. We have added a description of the preparation method of TEM samples in the experimental section.
Comments 10: The document cites relevant literature, but including recent studies on thermal stability in nanocrystalline alloys or other Al, systems would bring a more current context into play.
Response 10:
Thank you for the reviewer's suggestion. In fact, the introduction of the paper mentions the study of the stability of nanocrystalline alloys.
Comments 11: The resolution of the images could be enhanced to visualize grain size better.
Response 11:
Thank you for the reviewer's suggestion. We have further improved the resolution of TEM images.
Comments 12: It may be helpful to provide a quantitative assessment of the grain size observed in TEM images compared to the grain size generally simulated from X-ray diffraction.
Response 12:
Thank you for the reviewer's suggestion. We have added a grain size distribution map in the revised manuscript.
Comments 13: Enhance the contrast in the TEM images to make grain structures and boundaries more distinguishable.
Response 13:
Thank you for the reviewer's suggestions. We have made adjustments to the TEM images in the revised manuscript.
Comments 14: The effects of deposition parameters, such as working pressure, power, or substrate bias, on grain size must not be cheaper. They could affect the nanocrystalline structure significantly.
Response 14:
Thank you for the reviewer's suggestion. We have added descriptions of these deposition parameters in the experimental section.
Comments 15: In the case of magnetron sputtering, impurities from both the chamber and the targets can come into the system. Were some purity analyses done or steps taken to reduce contamination in the targets mentioned for these cases?
Response 15:
Thank you for the reviewer's suggestions. In terms of reducing pollution, we have adopted the following methods:
- Pump the vacuum degree of the chamber as high as possible;
- Use target materials with the highest possible purity;
- Clean the chamber as thoroughly as possible before each experiment.
Comments 16: Additional details, such as the deposition rate and process duration, are missing.
Response 16:
Thank you for the reviewer's suggestion. We have added relevant descriptions in the experimental section.

Reviewer 2 Report
Comments and Suggestions for Authors
Research on the strengthening mechanism of Al-Mo nanocrystalline alloy films based on the effects of various strengthening factors. The article is appropriately structured, with the methodology clearly described, specifying the preparation of the films through co-sputtering and the use of techniques such as XRD, TEM, EDS, and nanoindentation to study the microstructure and mechanical behaviours. Key results include the formation of a Mo solid solution in Al, Mo segregation at grain boundaries, a reduction in grain size, and an increase in hardness with higher Mo content. Finally, the conclusions regarding the strengthening mechanisms are implied, highlighting the contribution of grain size, solid solution, and nanocrystalline solute pinning. The introduction adequately addresses the background of the issue, mentioning the instability of nanocrystalline materials and how the addition of alloying elements, such as Mg in Al and Ta in Cu, can improve their stability, a topic that has been extensively investigated internationally. It also notes that the study of strengthening mechanisms in nanocrystalline alloys remains a challenge due to the complex interaction of microstructural factors, such as grain size, solute content, and grain boundary properties. At the end of the introduction, the study objectives are clearly defined: to synthesise Al-Mo alloy films through co-sputtering technology and analyse how varying Mo content affects the microstructure and mechanical properties, revealing the involved strengthening mechanisms.
Regarding the methodology, I have several suggestions, as more than 20 measurements were taken per sample to ensure the validity and representativeness of the results. Although specific details on the exact type of statistical processing are not provided, data analysis was performed using descriptive methods and the Oliver-Pharr method to calculate mechanical properties. Additionally, data validation was based on the use of calibrated instruments, such as EDS, XRD, TEM, and nanoindentation equipment, which are recognised and validated techniques for this type of analysis. However, further information could be included about the control of potential experimental biases, such as who performed the measurements and whether measures were taken to ensure consistency in the execution of the treatment, such as regular calibration of the equipment. It would also be useful to mention if any consensus method was employed in the interpretation of the results or validation of the findings, which could enhance the reliability of the obtained results. Finally, it is recommended to provide more details on the criteria for selecting Mo concentrations, clarifying whether other factors were considered besides the expected behaviour of alloys at different Mo contents.
Regarding the results, it is recommended to discuss in detail the Mo segregation at grain boundaries, particularly at concentrations exceeding 5.3 at.%, and its impact on the mechanical properties of the alloys. Furthermore, a deeper analysis of the three strengthening factors (grain refinement, solid solution strengthening, and nanocrystalline solute pinning) is necessary, highlighting their individual contributions to the alloy’s behaviour. The microstructural analysis using techniques such as XRD and TEM should be closely linked to the interpretation of the observed mechanical properties, explaining how crystal lattice distortion influences the characteristics of the alloy.
Author Response
Comments 1: Research on the strengthening mechanism of Al-Mo nanocrystalline alloy films based on the effects of various strengthening factors. The article is appropriately structured, with the methodology clearly described, specifying the preparation of the films through co-sputtering and the use of techniques such as XRD, TEM, EDS, and nanoindentation to study the microstructure and mechanical behaviours. Key results include the formation of a Mo solid solution in Al, Mo segregation at grain boundaries, a reduction in grain size, and an increase in hardness with higher Mo content. Finally, the conclusions regarding the strengthening mechanisms are implied, highlighting the contribution of grain size, solid solution, and nanocrystalline solute pinning. The introduction adequately addresses the background of the issue, mentioning the instability of nanocrystalline materials and how the addition of alloying elements, such as Mg in Al and Ta in Cu, can improve their stability, a topic that has been extensively investigated internationally. It also notes that the study of strengthening mechanisms in nanocrystalline alloys remains a challenge due to the complex interaction of microstructural factors, such as grain size, solute content, and grain boundary properties. At the end of the introduction, the study objectives are clearly defined: to synthesise Al-Mo alloy films through co-sputtering technology and analyse how varying Mo content affects the microstructure and mechanical properties, revealing the involved strengthening mechanisms.
Response 1:
Thank you for your positive evaluation of the manuscript.
Comments 2: Regarding the methodology, I have several suggestions, as more than 20 measurements were taken per sample to ensure the validity and representativeness of the results. Although specific details on the exact type of statistical processing are not provided, data analysis was performed using descriptive methods and the Oliver-Pharr method to calculate mechanical properties. Additionally, data validation was based on the use of calibrated instruments, such as EDS, XRD, TEM, and nanoindentation equipment, which are recognised and validated techniques for this type of analysis. However, further information could be included about the control of potential experimental biases, such as who performed the measurements and whether measures were taken to ensure consistency in the execution of the treatment, such as regular calibration of the equipment. It would also be useful to mention if any consensus method was employed in the interpretation of the results or validation of the findings, which could enhance the reliability of the obtained results. Finally, it is recommended to provide more details on the criteria for selecting Mo concentrations, clarifying whether other factors were considered besides the expected behaviour of alloys at different Mo contents.
Response 2:
Thank you for the reviewer's suggestions. The instruments and equipment used in the paper are from our university laboratory. The laboratory has passed CNAS certification. The instruments and equipment in the laboratory are regularly maintained and debugged to ensure the accuracy of experimental data.
The selection of Mo content range from 0 to 10.5 mainly has the following considerations: Based on the research results of this article, when the Mo content is below 10.5 at.%, the film forms a solid solution of Mo in Al. This simple structure is more suitable for studying and analyzing reinforcement mechanisms. When the Mo content exceeds 10.5 at.%, the film forms an amorphous structure, which exceeds the research scope of this paper.
Comments 3: Regarding the results, it is recommended to discuss in detail the Mo segregation at grain boundaries, particularly at concentrations exceeding 5.3 at.%, and its impact on the mechanical properties of the alloys. Furthermore, a deeper analysis of the three strengthening factors (grain refinement, solid solution strengthening, and nanocrystalline solute pinning) is necessary, highlighting their individual contributions to the alloy’s behaviour. The microstructural analysis using techniques such as XRD and TEM should be closely linked to the interpretation of the observed mechanical properties, explaining how crystal lattice distortion influences the characteristics of the alloy.
Response 3:
Thank you for the reviewer's suggestion. We have added relevant descriptions and analysis in the conclusion and discussion.

Reviewer 3 Report
Comments and Suggestions for Authors
This manuscript deals with with the synthesis and structural and mechanical properties evaluation of Al-Mo thin films.
The manuscript is well written and in general well discussed, however there are some issues that need to be attended.
1. In section 2.1 please use superscript whenever is needed.
2. Correct typos such as indentatio, a Al-Mo
3. In the same section authors confused Mo with Nb target.
4. In section 3.1 please use XRD pattern instead of XRD spectrum. Additionally, how would you explain an amorphous structure of the alloy?
5. From figure 2 discussion, What would be the physical interpretation of a linear fit of interplanar spacing vs Mon content? Why does the equation for d in the inset of fig 2b have GPa units?
6. In the abstract section, authors mention that part of the Mo segregates in the Al grain boundaries. However they do not show any evidence of this claim in XRD nor TEM results. What are the arguments to assume the mentioned Mo segregation?
7. Please briefly explain the physical origin of equation 1 and justify its use in the present work
Author Response
Comments 1: This manuscript deals with with the synthesis and structural and mechanical properties evaluation of Al-Mo thin films. The manuscript is well written and in general well discussed, however there are some issues that need to be attended.
Response 1:
Thank you for your positive evaluation of the manuscript.
Comments 2: In section 2.1 please use superscript whenever is needed.
Response 2:
Thank you for your suggestions. We have made revisions in the revised manuscript.
Comments 3: Correct typos such as indentatio, a Al-Mo.
Response 3:
Thank you for the reviewer's comments. We have carefully checked and revised the manuscript.
Comments 4: In the same section authors confused Mo with Nb target.
Response 4:
Thank you for pointing out the mistake. We have made corrections in the revised manuscript.
Comments 5: In section 3.1 please use XRD pattern instead of XRD spectrum. Additionally, how would you explain an amorphous structure of the alloy?
Response 5:
Thank you for your suggestions. We have made revisions in the revised manuscript. Regarding amorphous materials, extensive research has shown that the addition of alloying elements can cause lattice distortion in films, thereby damaging the integrity of the lattice. As the content of alloying elements increases, these lattice distortions gradually increase. The grain size of the film will rapidly decrease until it becomes amorphous.
Comments 6: From figure 2 discussion, What would be the physical interpretation of a linear fit of interplanar spacing vs Mon content? Why does the equation for d in the inset of fig 2b have GPa units?
Response 6:
Thank you for the reviewer's suggestion. The figure 2b shows the relationship between the interplanar spacing obtained from XRD results and the Mo content. The slope of the fitted curve will be used in the solid solution strengthening calculation formula. According to the Fleischer model, εb is the lattice mismatch coefficient, εb=(db/dc)/b.
Comments 7: In the abstract section, authors mention that part of the Mo segregates in the Al grain boundaries. However they do not show any evidence of this claim in XRD nor TEM results. What are the arguments to assume the mentioned Mo segregation?
Response 7:
The conclusion of Mo atom segregation was inferred from the results in Figure 4. The figure 4 shows the relationship between the total Mo content and the grain interior Mo. The total Mo content is measured by EDS, while the grain interior Mo content is calculated based on the interplanar spacing obtained from XRD. As shown in the figure, the grain interior Mo content is always slightly lower than the total Mo content. The difference between the two increases with the increase of Mo content. This indicates that when the content is low, Mo atoms are mainly solid dissolved in the lattice of Al. As the Mo content increases, a portion of Mo atoms will aggregate at the grain boundaries.
Comments 8: Please briefly explain the physical origin of equation 1 and justify its use in the present work.
Response 8:
The formula 1 is The empirical electron theory of solids and molecules, proposed by Yu Ruihuang in 1978, also known as the Yu theory (EET). For crystal structures with known lattice parameters, EET can provide the distribution of electrons on the bonding network and the state of atoms in the crystal, which can be used to calculate the binding energy, melting point, alloy phase diagram, etc. of the crystal. This formula can also be used to calculate the relationship between the lattice constant of a solid solution and the solute content. The calculation results have better applicability than Vegard's law.
